# Effects of a Chair Positioning Session on Awake Non-Intubated COVID-19 Pneumonia Patients: A Multicenter, Observational, and Pilot Study Using Lung Ultrasound

**DOI:** 10.3390/jcm11195885

**Published:** 2022-10-05

**Authors:** Alexandre Lopez, Pierre Simeone, Louis Delamarre, Gary Duclos, Charlotte Arbelot, Ines Lakbar, Bruno Pastene, Karine Bezulier, Samuel Dahan, Emilie Joffredo, Lucille Jay, Lionel Velly, Bernard Allaouchiche, Sami Hraiech, Marc Leone, Laurent Zieleskiewicz

**Affiliations:** 1Department of Anesthesiology and Intensive Care, Hôpital Nord, Assistance Publique Hôpitaux de Marseille, Aix-Marseille University, 13015 Marseille, France; 2Department of Anesthesiology and Critical Care Medicine, University Hospital Timone, AP-HM, Aix Marseille University, 13015 Marseille, France; 3CNRS, Inst Neurosci Timone, Aix Marseille University, UMR 7289, 13385 Marseille, France; 4Hospices Civils de Lyon, Centre Hospitalier Lyon-Sud, Service de Réanimation, 69310 Pierre-Bénite, France; 5Department of Anesthesiology and Critical Care Medicine, Faculté de Médecine et de Maïeutique Lyon Sud-Charles Mérieux, Université Claude Bernard Lyon 1, 69100 Villeurbanne, France; 6VetAgro Sup, Campus Vétérinaire de Lyon, 69007 Marcy l’Étoile, France; 7Service de Médecine Intensive—Réanimation, AP-HM, Hôpital Nord, 13015 Marseille, France; 8Health Service Research and Quality of Life Center (CEReSS), 27 Boulevard Jean-Moulin, Aix-Marseille Université, 13005 Marseille, France; 9Center for Cardiovascular and Nutrition Research (C2VN), Aix Marseille Université, INSERM, INRA, 13005 Marseille, France

**Keywords:** chair positioning, COVID-19 pneumonia, lung ultrasound, intensive care, acute respiratory failure

## Abstract

Background: LUS is a validated tool for the management of COVID-19 pneumonia. Chair positioning (CP) may have beneficial effects on oxygenation and lung aeration, and may be an easier alternative to PP. This study assessed the effects of a CP session on oxygenation and lung aeration (LA) changes in non-intubated COVID-19 patients. Methods: A retrospective multicenter study was conducted in an ICU. We analyzed data from LUS exams and SpO_2_:FiO_2_ performed before/after a CP session in non-intubated COVID-19 patients. Patients were divided into groups of responders or non-responders in terms of oxygenation or LA. Results: Thirty-three patients were included in the study; fourteen (44%) were oxygenation non-responders and eighteen (56%) were oxygenation responders, while thirteen (40.6%) and nineteen (59.4%) patients were classified as LA non-responders and responders, respectively. Changes in oxygenation and LA before/after a CP session were not correlated (r = −0.19, *p* = 0.3, 95% CI: −0.5–0.17). The reaeration scores did not differ between oxygenation responders and non-responders (1 (−0.75–3.75) vs. 4 (−1–6), *p* = 0.41). The LUS score was significantly correlated with SpO_2_:FiO_2_ before a CP session (r = 0.37, *p* = 0.04, 95% CI: 0.03–0.64) but not after (r = 0.17, *p* = 0.35, 95% CI: −0.19–0.50). Conclusion: A CP session was associated with improved oxygenation and LA in more than half of the non-intubated COVID-19 patients.

## 1. Introduction

Lung ultrasound (LUS) is a validated tool and possible alternative to chest computed tomography (CT) scans for the management of coronavirus disease 2019 (COVID-19)-related pneumonia and severity assessments [1,2]. Given its availability and feasibility, LUS could reduce the burden of overloaded CT departments. Furthermore, LUS is available at the bedside without requiring the transport of critically ill patients [3], thus reducing healthcare professionals’ risk of exposure to COVID-19 [4] and, in particular, patient-related adverse effects [5] during this pandemic. As a result, LUS is routinely used for triage and the management of intensive care unit (ICU) COVID-19 patients with acute respiratory failure. As prone positioning (PP) improves the outcomes of patients with moderate to severe acute respiratory distress syndrome (ARDS) [6], this positioning has been tested in non-intubated spontaneously breathing patients. In non-intubated COVID-19 patients, awake PP is associated with oxygenation improvement and a reduction in the need for mechanical ventilation [7]. However, the effect of PP is time-dependent, and the tolerance of prolonged awake PP is a major limitation of the procedure [8,9].

However, different positions, such as sitting/being upright, may have beneficial effects on respiratory mechanics and oxygenation in selected ARDS patients under (invasive) mechanical ventilation [10,11]. Therefore, chair positioning (CP) or Rodin’s Thinker CP could be an alternative to awake PP [12]. In addition, a recent study suggested an oxygenation improvement similar to that following awake PP after a Rodin’s Thinker positioning session [11]. The tolerability of CP may be better than that of PP in awake patients [11].

However, changes in positioning-related pathophysiological effects are always complex. During PP, LUS makes it possible to monitor lung aeration changes at the bedside [13,14]. A pilot study emphasized the potential role of LUS during awake PP [15].

Our study aimed to assess the effects of a CP session on oxygenation (using the pulse oxygen saturation:fractional inspired oxygen ratio (SpO_2_:FiO_2_)) and lung aeration (using lung reaeration scores) changes in non-intubated COVID-19 patients.

## 2. Materials and Methods

### 2.1. Study Design

A multicenter observational study was conducted in three intensive care units (ICUs) of French university hospitals (North Hospital of Marseille, La Timone Hospital Marseille, and South Hospital Lyon) using routine LUS examinations of COVID-19 patients. We retrospectively analyzed the collected data from LUS exams and the SpO_2_:FiO_2_ ratio performed before and after a CP session [13]. The study period was from January 2021 to April 2021. In our centers, a CP session was a common clinical use.

### 2.2. Ethical Considerations

The study protocol was approved by the Committee for Research Ethics of the French Society of Anesthesia and Intensive Care Medicine (IRB 00010254-2021-157). The patients received formal information on the use of their data. The different treatment strategies are considered standard care, and the analyses were performed retrospectively, so informed consent was waived according to French law [16].

### 2.3. Population

The COVID-19 patients admitted to ICUs with acute respiratory failure were included if they met the following criteria: (i) age of 18 or older, (ii) polymerase chain reaction (PCR)-documented severe acute respiratory syndrome coronavirus 2 (SARS-CoV-2) in nasopharyngeal samples upon ICU admission, (iii) spontaneous pulse oxygen saturation (breathing with no previous tracheal intubation during their ICU stay), and (iv) the need for oxygen to maintain SpO_2_ above 90%. A CP session was performed if patients presented all of the inclusion criteria in our standard care. The exclusion criteria were the presence of subcutaneous emphysema, a CP contraindication (altered vigilance, vomiting, hemoptysis, and spinal injury), need for immediate invasive mechanical ventilation upon ICU admission, symptomatic pericardial effusion, abdominal surgery (<1 month), pregnant or lactating woman, and a follow-up of less than 24 h.

### 2.4. Study Protocol

At ICU admission, each patient’s features (demographic and clinical history), Simplified Acute Physiology Score II (SAPS II) [17], and Sepsis-related Organ Failure Assessment (SOFA) score [18] were collected. Each patient’s SAPS II underwent a medical examination, arterial blood gas analysis, and monitoring of the heart rate, blood pressure, respiratory rate, and SpO_2_. For each patient, conventional oxygen therapy, high-flow nasal cannula therapy, or non-invasive ventilation were started to maintain SpO_2_ above 90%. In patients undergoing conventional oxygen therapy, the FiO_2_ was calculated as follows: FiO_2_ = (21 + 3 × oxygen flow (L/min))/100 [19]. The SpO_2_:FiO_2_ ratio was previously determined, and its use was correlated to the arterial pressure in oxygen (PaO_2_):FiO_2_ ratio in the respiratory component of the SOFA [20]. The SpO_2_:FiO_2_ ratio was collected according to the standard nurse monitoring protocol. In our standard of care, each patient underwent an LUS examination in a supine semi-recumbent position (constant 40 to 45°) at ICU admission, less than 10 min before CP (LUS_1_), and less than 10 min after CP (LUS_2_). Each eligible patient was installed on a medical armchair with the assistance of caregivers for at least three consecutive hours. The patients were positioned in a vertical trunk position. LUS was performed by imaging the 12 lung regions as described elsewhere [21]. To examine the posterior areas, the patients were placed in lateral decubitus. A 1–5 MHz convex probe with standard devices for ultrasound was used (Appendix A).

The regional aeration score featuring the anterior, lateral, and posterior lung areas was calculated. Points were allocated according to the worst ultrasound pattern observed. The LUS score corresponds to the sum of each examined area score (maximum score = 36) [22]. An ultrasound reaeration score was calculated as previously described [23] from the variation in the ultrasound pattern of each area examined between LUS_1_ and LUS_2_ (Appendix A). A positive reaeration score indicates an aeration gain, and a negative reaeration score indicates an aeration loss.

The LUS examinations were performed by senior (level 2 or 3) intensivists in charge of the patients [24,25]. The skill-levels of the operators were rated as follows [24]:-Level 2 operator: More than 25 supervised procedures and 200 non-supervised procedures.-Level 3 operator: LUS academic teacher with several publications in the field.

We collected the SpO_2_:FiO_2_ ratio at LUS_1_ and LUS_2_ [20]. Retrospectively, we defined responders and non-responders to CP sessions based on the reaeration score and SpO_2_:FiO_2_ ratio variations, respectively, as follows:-Oxygenation responders: Positive difference of SpO_2_:FiO_2_ ratios measured at LUS_2_ and LUS_1_.-Oxygenation non-responders: Negative difference of SpO_2_:FiO_2_ ratios measured at LUS_2_ and LUS_1_.-Lung aeration responders: A positive reaeration score between LUS_2_ and LUS_1_.-Lung aeration non-responders: A negative reaeration score between LUS_2_ and LUS_1_.

### 2.5. Outcomes

Our study aimed to assess the effects of a CP session on oxygenation (using the SpO_2_:FiO_2_ ratio) and lung aeration (using lung reaeration scores) changes in non-intubated COVID-19 patients.

The secondary objectives were to assess the following:-The mechanisms of change in oxygenation during a CP session are reflected by the correlation between oxygenation and lung aeration changes.-The performance of baseline LUS can be used to predict the effect of a CP session on oxygenation and lung aeration.-The effect of oxygenation and lung aeration responses on the outcomes (need for invasive mechanical ventilation, length of stay, ICU mortality, etc.).

### 2.6. Statistical Analysis

No sample size was calculated a priori, given the exploratory nature of our study. Patients were included based on convenience sampling. Quantitative variables are expressed in median and interquartile ranges. Qualitative variables are expressed in numbers and percentages. Missing values were omitted from the analysis, as no data imputation was planned. The primary outcome was assessed using a comparison of the lung aeration index (total LUS score) between the two time points with a non-parametric Mann–Whitney U test. The secondary outcomes were assessed using the same test. The correlations between quantitative variables were studied via Pearson’s correlation coefficient, its *p*-value, and the 95% confidence interval (CI). The quantitative variables were compared between groups using a non-parametric Mann–Whitney U test. The qualitative variables were compared between groups using Fisher’s exact test or a chi-squared test in cases of more than two classes. This observational study was reported in compliance with the STROBE guidelines [26]. All the comparisons were two-tailed. A *p*-value < 0.05 was required for statistical significance. Data analysis was performed using R software (Version: R 4.1.2. Vienna, Austria) [27].

## 3. Results

### 3.1. Enrolled Patients’ Characteristics

The final analysis was performed on a population of 33 patients (Figure 1). The patients’ features are illustrated in Table 1. Our cohort consisted of 58% males (42% females) with a median age of 67 (range: 53–74) years with a SAPS II of 29 (range: 24–32). Twenty-five (76%) patients received oxygen via a high-flow nasal cannula, and the median SpO_2_:FiO_2_ rate was 140 (range: 111–195).

### 3.2. Primary Outcome

Based on the SpO_2_:FiO_2_ ratio variations, 14 (44%) oxygenation non-responders and 18 (56%) oxygenation responders were identified. The SpO_2_:FiO_2_ ratios were 140 (range: 110–190) before CP and 160 (range: 120–200) after CP (*p* = 0.4) (Figure 2). Based on the reaeration score, 13 (40.6%) and 19 (59.4%) patients were classified as lung aeration non-responders and responders, respectively. The median reaeration score was 2 (interquartile range (IQR): −2, 6), with a median reaeration score of −1 (IQR: −5, −1) in non-responders and 5 (IQR: 3, 7) in responders (Table 1). The individual values of oxygenation and lung aeration changes during a CP session are presented in Figure 3. One missing value in the reaeration score explains the discrepancy in the final analysis of 32 patients.

### 3.3. Secondary Outcomes

The changes in lung aeration and oxygenation (defined by the SpO_2_:FiO_2_ ratio) before and after a CP session were not correlated. The reaeration scores did not differ between responders and non-responders in terms of oxygenation (1 (−0.75–3.75) vs. 4 (−1–6), *p* = 0.41). The LUS score was significantly correlated with the SpO_2_:FiO_2_ ratio before a CP session (Pearson’s r = 0.37, *p* = 0.04, 95% CI: 0.03–0.64) but not after (Pearson’s r = 0.17, *p* = 0.35, 95% CI: −0.19–0.50), as shown in Figure 3a,b. Oxygenation changes and reaeration scores after the CP sessions were not correlated (Pearson’s r = −0.19, *p* = 0.3, 95% CI: −0.5–0.17) (Figure 3c).

In terms of predicting a CP session response, the baseline LUS score was similar in oxygenation responders and non-responders (median: 20, IQR: (17–24) vs. 22 (20–23), *p* = 0.8) (Appendix A). The changes in the SpO_2_:FiO_2_ ratio did not significantly differ in patients with or without consolidations before the CP session (0.02, (−0.01–0.2) vs. 0.04 (−0.01–0.1), *p* = 0.9) (Appendix A). Similarly, the baseline LUS score was similar in the lung aeration responders and non-responders (median: 22, IQR: (20–24) vs. 19 (18–23), respectively, *p* = 0.2) (Appendix A). The reaeration score did not differ between patients with and without consolidations before the CP session (0.5, (−1.0–6.0) vs. 2.5 (−0.5–4.75), respectively, *p* = 0.09) (Appendix A). Regional LUS scores before and after the CP session are presented in Appendix A.

The outcomes, including the need for invasive mechanical ventilation, duration of ICU and hospital stay, and mortality rates in the ICU, in hospital, and at day 28, were similar between the responders and non-responders in terms of the oxygenation (Appendix A) and lung aeration responses (Appendix A).

## 4. Discussion

Our study showed that a CP session was associated with an improvement in oxygenation and pulmonary aeration in more than half of our COVID-19 patients. However, the results did not show a significant correlation between oxygenation and lung aeration changes.

To the best of our knowledge, this study is the first to report the effects of a CP session on oxygenation and lung aeration in awake COVID-19 patients. A single-center study including 25 patients found that Rodin’s Thinker positioning was associated with a significant subsequent improvement in oxygenation of more than 40 mmHg of PaO_2_ measured immediately at the end of the session [12]. Rodin’s Thinker positioning may be theoretically more efficient than classical CP because it reverses the gravitational gradient in a manner similar to PP. Nevertheless, the mechanism underlying this effect was not assessed in this study. Furthermore, the mean PaCO_2_ was not altered after a Rodin’s Thinker positioning session, suggesting an insignificant effect on lung aeration. To summarize, CP and Rodin’s Thinker positioning are feasible and may improve oxygenation in selected patients, but larger studies are needed to confirm these findings.

The pathophysiological effects of positioning in patients with acute respiratory failure are difficult to assess [28].

Recently, Giosa et al. [29] found that orthodeoxia is frequent in COVID-19 patients and might partly explain the benefits of awake prone position. We cannot exclude that this phenomenon was present in the oxygenation non-responder patients of our cohort. The use of LUS allows for the bedside assessment of regional lung aeration. Thus, it represents a useful tool for assessing changes in patients with acute respiratory failure [13]. The LUS reaeration score was recently validated in comparison to a gold standard method—end-expiratory lung volume measured by an automated nitrogen washout/washing technique—for PP-induced lung inflation [14]. In a previous study, we assessed regional aeration changes during a PP session in 51 ARDS patients [13]. We found that changes in aeration and oxygenation were not correlated, suggesting that both ventilation and perfusion were critical determinants of oxygenation. In fact, using a different tool, our current results are in line with several physiopathological recent studies on awake PP effects during COVID-19. In a recent case report on a COVID-19 patient on PP using thoracic Electrical Impedance Tomography (EIT), Zarantonello et al. concluded that oxygenation gain was related to ventilation:perfusion matching improvement [30]. Combining a CT scan and EIT, Fossali et al. showed that PP effects on COVID-19 patients are mainly driven by improved ventilation:perfusion matching [31]. Finally, using EIT and an intrapulmonary shunt calculation (based on a modified Berggren equation), a recent study confirmed that PP is associated with a reduction in an intrapulmonary shunt [32]. Thus, the pulmonary blood flow could be diverted away from the reaerated lung regions in PP, resulting in ventilation:perfusion matching alteration [33].

Similar to PP, CP affects the gravitational gradient, end-expiratory lung volume, and hemodynamics. However, the magnitude of these changes is poorly described. In our patients, the lack of a correlation between oxygenation and lung aeration changes also suggests ventilation:perfusion matching alteration during a CP session, as reported during PP in mechanically ventilated COVID-19 patients [34]. It is worth noting that Langer et al. [35] showed that, in patients under invasive MV, PP was associated with improved oxygenation without any modification of respiratory system compliance, suggesting that lung recruitment was not the major mechanism. Previous studies using the ultrasound monitoring of lung aeration during PP found similar results [13,14]. In our study, lung involvement morphology assessed with baseline LUS scores did not predict subsequent changes in lung aeration or oxygenation. Only limited previous data have suggested that lung involvement morphology could predict oxygenation responses to PP in intubated or non-intubated patients [15,36]. However, larger studies have shown that baseline lung involvement profiles, either evaluated with LUS or chest computed tomography, were not predictive of responses to PP [13,37]. Thus, it seems that the complex mechanisms induced by changes in positioning are not predictable with the assessment of baseline lung involvement morphology, regardless of the imaging technique. Even if less helpful in clinical practice, changes in LUS scores during the early phase of a PP session could more accurately predict the PP response and outcome [14,38]. Future studies should determine if an early response to a CP session is predictive of oxygenation, lung aeration, or outcome improvement.

Our study has several limitations. As a retrospective analysis, we included only a convenience sample of patients who met the inclusion criteria. Due to its observational nature, we did not perform a systematic blood gas analysis at different time points. As in many other studies, we used the SpO_2_:FiO_2_ ratio is a validated surrogate of the PaO_2_:FiO_2_ ratio [20]. However, both PaO_2_:FiO_2_ and SpO_2_:FiO_2_ ratio may have intrinsic limitations during COVID-19 pneumonia management [39,40]. As our study was non-interventional, we did not perform any systematic chest CT scan. Thus, not all of the patients had a chest CT scan, and, more importantly, the delay between the chest CT scan and CP was highly variable. Consequently, we could not compare the LUS examination with chest CT scan. The number of patients was relatively small, and we did not include a control group, which precluded the analysis of clinical outcomes. In our population, the median delay between hospital admission and CP session in ICU was 8 days indicating a well-established pneumonia which may have impacted our results. Thus, the effect of a CP session on the outcome, including the need for invasive ventilation, duration of ICU and hospital stay, and mortality rate, remains to be assessed with further well-designed studies. However, this multicenter study reflects a global approach used by different teams. Finally, we included only non-intubated patients at the onset of their ICU stays. Therefore, as the response to PP and recruitment decreases over time due to progressive lung consolidations versus atelectasis [34], we cannot expand our results to patients previously intubated or those in a late stage of their disease progression.

## 5. Conclusions

In this pilot study, a CP session was found to be associated with improved oxygenation and lung aeration in more than half of the non-intubated COVID-19 patients. However, oxygenation and lung aeration changes were not associated, suggesting that a CP session induces ventilation:perfusion matching alteration. These preliminary results suggest that a CP session may be associated with improved oxygenation and lung aeration in select patients. The prediction of CP session responses and their effects on outcomes need to be addressed in future larger studies.

## Figures and Tables

**Figure 1 jcm-11-05885-f001:**
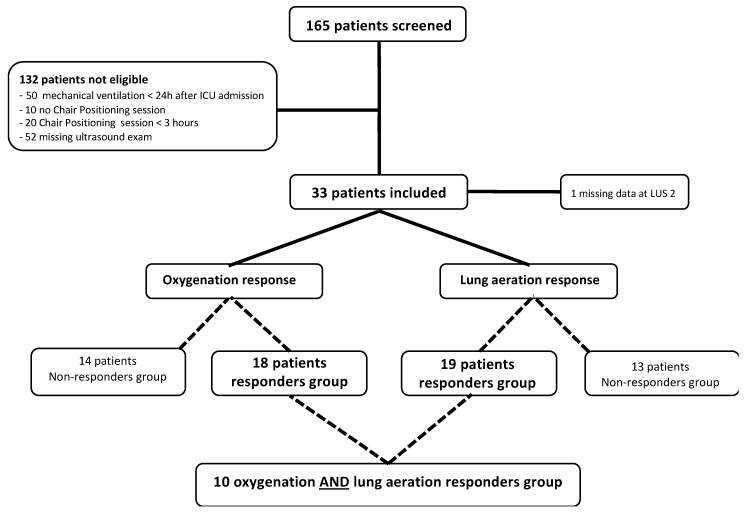
Flow chart of the study.

**Figure 2 jcm-11-05885-f002:**
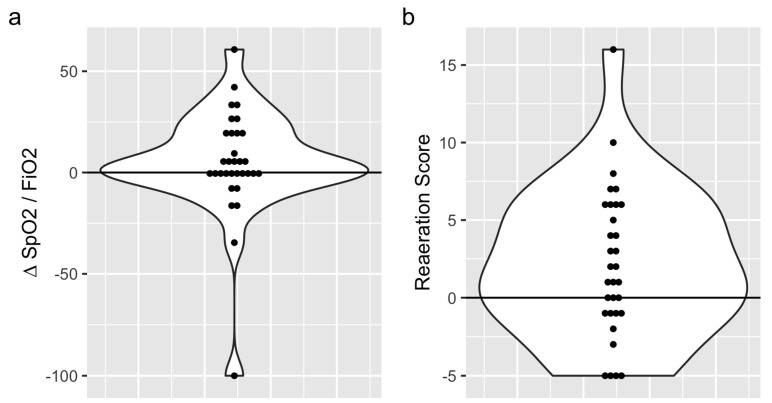
Variations in the SpO_2_:FiO_2_ ration (**a**) and reaeration score (**b**) after the chair trial in the whole cohort.

**Figure 3 jcm-11-05885-f003:**
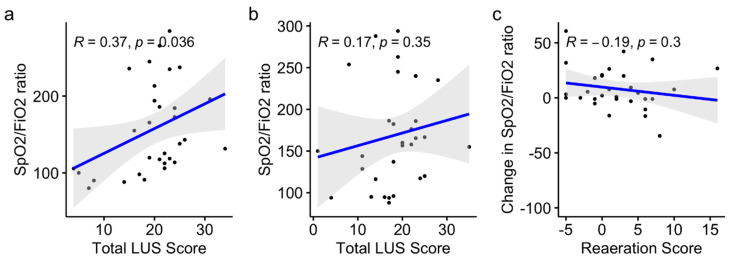
Variations in the SpO_2_:FiO_2_ ratio and reaeration score before (**a**) and after (**b**) the CP session, and variation of oxygenation and reaeration after CP session (**c**) in the whole cohort.

**Table 1 jcm-11-05885-t001:** Baseline clinical data of patients in the whole cohort and according to their response to chair positioning in terms of oxygenation and lung aeration.

		In Terms of Oxygenation			In Terms of Lung Aeration		
	Whole Cohort (*n* = 33)	Non-Responders Group (*n* = 14)	Responders Group (*n* = 18)	*p*-Value	Non-Responders Group (*n* = 13)	Responders Group (*n* = 19)	*p*-Value
Demographic Data							
Sex, *n* (%)							
Male	19 (58)	6 (43)	12 (67)	0.3	8 (62)	11 (58)	1
Female	14 (42)	8 (57)	6 (33)		5 (38)	8 (42)	
Age, median (IQR)	67 (53–74)	67 (60–71)	66 (60–71)	0.7	73 (55–75)	66 (50–69)	0.8
BMI, median (IQR)	30 (25–33)	29 (24–31)	32 (29–37)	0.01	30 (28–33)	28 (24–33)	0.3
**Comorbidities**							
Cancer, *n* (%)	3 (9)	0	3 (17)	0.6	1 (8)	2 (11)	1
COPD, *n* (%)	2 (6)	0	2 (11)	0.6	1 (8)	1 (5)	1
Coronary disease, *n* (%)	1 (3)	0	1 (6)	1	1 (8)	0 (0)	0.8
Diabetes, *n* (%)	8 (24)	2 (14)	5 (28)	0.6	6 (46)	2 (11)	0.06
Hypertension, *n* (%)	14 (42)	5 (36)	8 (44)	0.7	7 (54)	7 (37)	0.5
Immunodeficiency ^a^, *n* (%)	1 (3)	1 (7)	0	0.9	1 (8)	0 (0)	0.8
Chronic kidney disease, *n* (%)	0	1 (7)	0	0.4	1 (8)	0 (0)	0.2
History of stroke, *n* (%)	2 (6)	1 (7)	1 (6)	1	1 (8)	1 (5)	1
Tobacco consumption, *n* (%)	8 (24)	3 (21)	5 (28)	0.9	2 (15)	6 (32)	0.6
**Clinical Data at Admission**							
SAPS II ^b^, median (IQR)	29 (24–32)	28 (24–31)	30 (23–36)	0.6	31 (24–41)	28 (22–30)	0.2
SOFA, median (IQR)	3 (3–3)	3 (3–3)	3 (3–4)	0.1	3 (3–4)	3 (3–4)	0.6
SpO_2_, median (IQR), %	93 (92–96)	93 (93–96)	93 (92–95)	0.5	93 (91–93)	94 (93–97)	0.06
FiO_2_, median (IQR), %	100 (50–100)	100 (79–100)	100 (50–100)	0.6	100 (85–100)	100 (55–100)	0.5
Respiratory rate, median (IQR)	28 (22–35)	25 (22–28)	30 (21–38)	0.9	24 (22–30)	30 (25–38)	0.3
MAP, median (IQR), mmHg	86 (81–92)	90 (84–90)	84 (81–95)	0.6	79 (77–102)	88 (84–91)	0.8
Heart rate, median (IQR), bpm	89 (73–99)	80 (69–91)	98 (77–105)	0.3	84 (67–103)	890 (77–97)	0.9
Thrombosis, *n* (%)	19 (58)	1 (7)	1 (6)	1	1 (8)	0 (0)	0.9
Noradrenaline use, *n* (%)	8 (24)	5 (36)	3 (17)	0.9	5 (39)	2 (11)	0.2
**Respiratory Support at Admission**							
HFNC use, *n* (%)	25 (76)	12 (86)	12 (67)	1	9 (69)	15 (79)	0.2
HFNC flow, median (IQR), L/min	50 (12–50)	45 (18–50)	35 (11–50.0)	0.9	50 (20–50)	40 (10–50)	0.7
**Clinical and Ultrasound Data at LUS1**							
Duration between admission and LUS1, median (IQR), days	8 (6–10)	8 (5–8)	10 (6–11)	0.3	7 (5–9)	8 (6–13)	0.09
SpO_2_, median (IQR), %	94 (91–95)	93 (91–94)	94.0 (92–96)	0.7	93 (91–95)	94 (93–95)	0.2
FiO_2_, median (IQR), %	90 (55–100)	95 (69–100)	73 (50–100)	0.3	100 (55–100)	85 (53–100)	0.9
SpO_2_/FiO_2_ ratio, median (IQR), %	140 (111–195)	131 (107–184)	146 (115–209)	0.8	135 (112–171)	173 (113–205)	0.3
Respiratory rate, median (IQR)	22 (21–25)	22 (22–24)	22 (20–27)	0.9	22 (19–24)	22 (21–26)	0.2
Heart rate, median (IQR), bpm	83 (72–92)	86 (72–98)	82 (73–90)	0.9	76 (68–90)	85 (74–96)	0.5
MAP, median (IQR), mmHg	88 (79–90)	88 (77–90)	87 (80–92)	0.5	84 (79–87)	88 (78–90)	0.8
HFNC flow, median (IQR), L/min	50 (20–50)	50.0 (20.0–50.0)	50.0 (30.0–50)	0.6	50 (30–50)	50 (20–50)	0.6
Total LUS score, median (IQR)	21 (18–24)	22 (20–23)	20 (17–24)	0.8	19 (18–23)	22 (20–24)	0.7
Right-sided consolidation, *n* (%)	15 (46)	5 (36)	10 (56)	0.4	7 (37)	15 (46)	0.3
Left-sided consolidation, *n* (%)	16 (49)	8 (57)	8 (44)	0.7	8 (62)	8 (42)	0.4
Right-sided pleural effusion, *n* (%)	2 (6)	2 (14)	0 (0)	0.4	1 (8)	1 (5)	1
Left-sided pleural effusion, *n* (%)	2 (6)	2 (14)	0 (0)	0.4	1 (8)	1 (5)	1
**Clinical and Ultrasound Data at LUS2**							
s	95 (93–96)	93 (91–95)	95 (93–96)	0.09	95 (91–96)	94 (93–96)	0.7
FiO_2_, median (IQR), %	88 (59–100)	100 (80–100)	70 (53–100)	0.2	70 (60–100)	95 (55–100)	0.8
SpO_2_:FiO_2_ ratio, median (IQR), %	159 (119–198)	120 (95–182)	163 (145–240)	0.04	158 (129–167)	182 (118–238)	0.6
Respiratory rate, median (IQR), bpm	24 (21–28)	24 (23–26)	23 (21–29)	0.9	23 (22–25)	24 (21–28)	0.3
Heart rate, median (IQR), bpm	77 (68–84)	74 (65–81)	77 (69–85)	0.3	78 (74–82)	75 (67–84)	0.5
MAP, median (IQR), mmHg	85 (74–90)	74 (72–95)	85 (83–90)	0.6	86 (84–90)	78 (72–94)	0.2
HFNC flow, median (IQR), L/min	50 (14–50)	40 (20–50)	50 (13–50)	0.9	50 (20–50)	50 (13–50)	0.9
Total LUS score, median (IQR)	19 (16–23)	18 (17–22)	19.0 (11.8–23)	0.8	22 (19–23)	17 (14–19)	0.02
Right-sided consolidation, *n* (%)	10 (30)	3 (21.4%)	7 (38.9%)	0.6	6 (46)	4 (21)	0.2
Left-sided consolidation, *n* (%)	13 (39)	4 (28.6%)	9 (50.0%)	0.5	7 (54)	6 (31)	0.4
Right-sided pleural effusion, *n* (%)	4 (12)	2 (14.3%)	2 (11.1%)	0.3	2 (15)	2 (11)	1
Left-sided pleural effusion, *n* (%)	2 (6)	2 (14.3%)	0 (0%)	0.3	1 (8)	1 (5)	1
Global reaeration score, median (IQR)	2 (−1; 6)	4.00 (−1; 6)	1 (−0.7; 4)	0.6	−1 (−5; −1)	5 (3–7)	<0.001
∆SpO_2_:FiO_2_ ratio, median (IQR), %	0.3 (−0.01; 0.2)	−0.02 (−0.1; −0.01)	0.19 (0.06–0.3)	<0.001	0.04 (0–0.2)	0.01 (−0.02; 0.1)	0.3

Abbreviations: BMI, body mass index; COPD, chronic obstructive pulmonary disease; HFNC, high-flow nasal canula; SpO_2_:FiO_2,_ ratio of pulse oximetry to fractional inspired oxygen; LUS, lung ultrasound; MAP, mean arterial pressure; SAPS II, Simplified Acute Physiology Score II; SOFA, Sepsis-related Organ Failure Assessment, and IQR, interquartile range. ^a^ Immunodeficiency: HIV patients, transplant patients, and patients undergoing immunosuppressive treatments. ^b^ The SAPS II ranges from 0 to 163, with higher scores indicating a higher risk of mortality. A patient with a score of 30 has an estimated mortality risk of 10%.

## Data Availability

The data presented in this study are available on request from the corresponding authors: alexandre.lopez@ap-hm.fr (A.L.); pierre.simeone@ap-hm.fr (P.S.); marc.leone@ap-hm.fr (M.L.); laurent.zieleskiewicz@ap-hm.fr (L.Z.).

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
