# Peer review of "Effects of a Chair Positioning Session on Awake Non-Intubated COVID-19 Pneumonia Patients: A Multicenter, Observational, and Pilot Study Using Lung Ultrasound"

_jcm, 2022, doi:10.3390/jcm11195885_

Round 1
Reviewer 1 Report
Dear Authors,
there are some issues to address.
- English language should be revised, including punctuation.
- Page 2 line 47: this point could be clearer by highlighting both the risk for the patients and for healthcare personnel.
- Please better specify what “senior level-2 or 3 intensivists” means, for example in terms of years in service.
- Discussion of results and conclusions should highlight that “the outcomes, including the need for invasive mechanical ventilation, duration of 225 ICU and hospital stay, and mortality rates in the ICU, hospital, and at day 28 were similar between the responders and non-responders in terms of oxygenation”. This is important because no impact on important outcomes is ascribable to a single session of CP, according to this study’s results.
Author Response
RESPONSE TO REFEREE #1
Dear Authors,
There are some issues to address.
Point 1: English language should be revised, including punctuation.
R : Thank you.
According to your demand, we performed another English editing on MDPI English editing. We add the copy of English editing certificate.
Point 2: Page 2 line 47: this point could be clearer by highlighting both the risk for the patients and for healthcare personnel.
R: Thank you for this advice.
We corrected in the manuscript accordingly.
« Lung ultrasound (LUS) is a validated tool and possible alternative to chest computed tomography (CT) scans for the management of coronavirus disease 2019 (COVID-19)-related pneumonia and severity assessments [1][2]. Given its availability and feasibility, LUS could reduce the burden of overloaded CT departments. Furthermore, LUS is available at the bedside without requiring the transport of critically ill patients [3], thus reducing healthcare professionals’ risk of exposure to COVID-19 [4] and, in particular, patient-related adverse effects [5] during this pandemic. »
Point 3: Please better specify what “senior level-2 or 3 intensivists” means, for example in terms of years in service.
Thank you for this remark
We modified accordingly in the manuscript
« The LUS examinations were performed by senior (level 2 or 3) intensivists in charge of the patients [24][25]. The skill of the operators was rated as follows [24]:
- Level 2 operator: More than 25 supervised procedures and 200 non-supervised procedures.
- Level 3 operator: LUS academic teacher with several publications in the field »
Point 4: Discussion of results and conclusions should highlight that “the outcomes, including the need for invasive mechanical ventilation, duration of 225 ICU and hospital stay, and mortality rates in the ICU, hospital, and at day 28 were similar between the responders and non-responders in terms of oxygenation”. This is important because no impact on important outcomes is ascribable to a single session of CP, according to this study’s results.
Thank you for this comment
According to your advice, we added a comment in the limitation section (L 288):
« Thus, the effect of CP session on the outcome ; including the need for invasive ventilation, duration of ICU and hospital stay and mortality rate ; remains to be assessed with further well-designed studies. »

Reviewer 2 Report
This retrospective study assessed the effects of a chair positioning session on oxygenation and lung aeration changes in non-intubated COVID-19 patients. Lung ultrasound and SpO2/FiO2 ratio were analyzed. Chair positioning session was associated with improved oxygenation and lung aeration in more than half of the non-intubated COVID-19 patients. It is advisable to better discuss pathophysiological background and meaning of the obtained results. Computed tomography data is not presented in th manuscript. Reference list should be updated.
It is highly advisable to expand this study and run it as prospective RCT.
Author Response
RESPONSE TO THE REVIEW
ORIGINAL ARTICLE
Title : Effects of a Chair Positioning Session on Awake Non-Intubated COVID-19 Pneumonia Patients: A Multicenter, Observational, and Pilot Study Using Lung Ultrasound
Dear Editor
Thank you for your confidence and pertinent advice.
We provide here by a point-by-point answer.
Kind regards
RESPONSE TO REFEREE #2
Point 1 : This retrospective study assessed the effects of a chair positioning session on oxygenation and lung aeration changes in non-intubated COVID-19 patients. Lung ultrasound and SpO2/FiO2 ratio were analyzed. Chair positioning session was associated with improved oxygenation and lung aeration in more than half of the non-intubated COVID-19 patients. It is advisable to better discuss pathophysiological background and meaning of the obtained results.
Thank you for this advice. Pathophysiological background was already extensively discussed in our manuscript.
However, according to your demand, we added new recent references describing the same physiopathological hypothesis during changes in positioning in COVID-19 patients:
« In fact, using a different tool, our current results are in line with several physiopathological recent studies on awake PP effects during COVID-19. In a recent case report on a COVID-19 patient on PP using thoracic Electrical Impedance Tomography (EIT), Zarantonello et al. concluded that oxygenation gain was related to a ventilation:perfusion matching improvement [30]. Combining CT scan and EIT, Fossali et al. showed that PP effects in COVID-19 patients are mainly driven by improved ventilation:perfusion matching [31]. Finally, using EIT and intrapulmonary shunt calculation (based on modified Berggren equation) a recent study confirmed that PP is associated to a reduction in intrapulmonary shunt [32]. Thus, during changes in position, in covid 19 patients, the pulmonary blood flow could be diverted away from the reaerated lung regions, resulting in ventilation:perfusion matching alteration »
Point 2: Computed tomography data is not presented in the manuscript
Thank you for this comment.
As our study was non interventional, we did not perform any systematic chest CT scan. Thus not all the patient had CT scan and more importantly the delay between chest CT scan and Chair positioning session was highly variable.
We added this information in the limitation section as follow:
« As our study was non-interventional, we did not perform any systematic chest CT scan. Thus not all of the patients had chest CT scan, and, more importantly, the delay between chest CT scan and CP was highly variable »
Point 3: Reference list should be updated.
Thank you for this remark.
As per your advice, we have corrected the list of references. We have also performed a new analysis of the current literature on the subject and added the reference.
- Zarantonello, F.; Andreatta, G.; Sella, N.; Navalesi, P. Prone Position and Lung Ventilation and Perfusion Matching in Acute Respiratory Failure Due to COVID-19. Am J Respir Crit Care Med 2020, 202, 278–279, doi:10.1164/rccm.202003-0775IM.
- Fossali, T.; Pavlovsky, B.; Ottolina, D.; Colombo, R.; Basile, M.C.; Castelli, A.; Rech, R.; Borghi, B.; Ianniello, A.; Flor, N.; et al. Effects of Prone Position on Lung Recruitment and Ventilation-Perfusion Matching in Patients With COVID-19 Acute Respiratory Distress Syndrome: A Combined CT Scan/Electrical Impedance Tomography Study. Crit Care Med 2022, 50, 723–732, doi:10.1097/CCM.0000000000005450.
- Dos Santos Rocha, A.; Diaper, J.; Balogh, A.L.; Marti, C.; Grosgurin, O.; Habre, W.; Peták, F.; Südy, R. Effect of Body Position on the Redistribution of Regional Lung Aeration during Invasive and Non-Invasive Ventilation of COVID-19 Patients. Sci Rep 2022, 12, 11085, doi:10.1038/s41598-022-15122-9.
- Langer, T.; Brioni, M.; Guzzardella, A.; Carlesso, E.; Cabrini, L.; Castelli, G.; Dalla Corte, F.; De Robertis, E.; Favarato, M.; Forastieri, A.; et al. Prone Position in Intubated, Mechanically Ventilated Patients with COVID-19: A Multi-Centric Study of More than 1000 Patients. Crit Care 2021, 25, 128, doi:10.1186/s13054-021-03552-2
Point 4: It is highly advisable to expand this study and run it as prospective RCT.
You are right as underlined in the conclusion
«The prediction of CP session response and its effects on outcomes need to be addressed in future larger studies

Round 2
Reviewer 1 Report
Dear Authors,
thank you for your replies.
Author Response
Dear Reviewer,
We thank you for this reviewing.
The Authors